# Assessment of Sexual Dysfunction in Cervical Cancer Patients after Different Treatment Modality: A Systematic Review

**DOI:** 10.3390/medicina58091223

**Published:** 2022-09-05

**Authors:** Francesco Tramacere, Valentina Lancellotta, Calogero Casà, Bruno Fionda, Patrizia Cornacchione, Ciro Mazzarella, Rosa Pasqualina De Vincenzo, Gabriella Macchia, Martina Ferioli, Angeles Rovirosa, Maria Antonietta Gambacorta, Cesare Colosimo, Vincenzo Valentini, Roberto Iezzi, Luca Tagliaferri

**Affiliations:** 1S.C. Radioterapia, ASLBR Ospedale “A. Perrino” Brindisi, 72100 Brindisi, Italy; 2UOC Radioterapia Oncologica, Dipartimento di Diagnostica per Immagini, Radioterapia Oncologica ed Ematologia, Fondazione Policlinico Universitario “A. Gemelli” IRCCS, 00128 Rome, Italy; 3Department of Woman and Child Health and Public Health, Woman Health Area, Fondazione Policlinico Universitario A. Gemelli IRCCS, 00128 Rome, Italy; 4Radiation Oncology Unit, Gemelli Molise Hospital, Università Cattolica del Sacro Cuore, 86100 Campobasso, Italy; 5Radiation Oncology Center, Department of Experimental, Diagnostic and Specialty Medicine-DIMES, University of Bologna, 40138 Bologna, Italy; 6Department of Radiation Oncology, Hospital Clinic Barcelona, 08036 Barcelona, Spain; 7Fonaments Clinics Department, Faculty of Medicine, Universitat de Barcelona, 08036 Barcelona, Spain; 8Istituto di Radiologia, Università Cattolica del Sacro Cuore, 00128 Rome, Italy; 9Department of Radiological Sciences, School of Medicine, Catholic University, 00128 Rome, Italy

**Keywords:** cervix cancer, interventional radiotherapy (brachytherapy), sexual dysfunction, radio-chemotherapy

## Abstract

*Background and Objectives*: Cervical cancer is a leading cause of mortality among women. Chemo-radiation followed by interventional radiotherapy (IRT) is the standard of care for stage IB–IVA FIGO. Several studies have shown that image-guided adaptive IRT resulted in excellent local and pelvic control, but it is associated with vaginal toxicity and intercourse problems. The purpose of this review is to evaluate the dysfunctions of the sexual sphere in patients with cervical cancer undergoing different cervix cancer treatments. *Materials and Methods*: We performed a comprehensive literature search using Pub med, Scopus and Cochrane to identify all the full articles evaluating the dysfunctions of the sexual sphere. ClinicalTrials.gov was searched for ongoing or recently completed trials, and PROSPERO was searched for ongoing or recently completed systematic reviews. *Results*: One thousand three hundred fifty-six women included in five studies published from 2016 to 2022 were analyzed. The median age was 50 years (range 46–56 years). The median follow-up was 12 months (range 0–60). Cervical cancer diagnosis and treatment (radiotherapy, chemotherapy and surgery) negatively affected sexual intercourse. Sexual symptoms such as fibrosis, strictures, decreased elasticity and depth and mucosal atrophy promote sexual dysfunction by causing frigidity, lack of lubrication, arousal, orgasm and libido and dyspareunia. *Conclusions*: Physical, physiological and social factors all contribute to the modification of the sexual sphere. Cervical cancer survivors who were irradiated have lower sexual and vaginal function than the normal population. Although there are cures for reducing discomfort, effective communication about sexual dysfunctions following treatment is essential.

## 1. Introduction

Cervical cancer is a leading cause of mortality among women [1]. In 2020, an estimated 604,000 women were diagnosed with cervical cancer worldwide and about 342,000 women died from the disease [1].

The standard of care for stage IB–IVA FIGO (International Federation of Gynecology and Obstetrics) consists of chemo-radiation (weekly intravenous cisplatin 40 mg/m^2^, 5–6 cycles, 1 day per cycle, plus 45–50 Gy external-beam radiotherapy (EBRT) delivered in 1.8–2 Gy fractions) followed by image-guided Interventional Radiotherapy (IG-IRT, also called brachytherapy) [2].

Several studies have shown that IG-IRT resulted in excellent local and pelvic control [3,4,5]. The 3-year actuarial pelvic control rate was 96% for stage IB disease, 89% for stage IIB disease, and 73% for stage IIIB disease. At the same time, major morbidity (grades 3–5) was limited after IG-IRT (3–6% per organ). As a result of all of this, the number of long-term survivors has increased and, in recent years, research has focused on the quality of life of these patients, with sexual health being one of the most essential parts of life. Vaginal morbidity is a documented side effect of pelvic radiation therapy. The vaginal wall is sensitive to the effects of radiation and is especially important in the treatment of cervical cancer considering the age of the patients, such as when EBRT, IRT, and chemotherapy (CT) are combined [6]. Sexual health and orgasm, however, encompass more than the sum of vaginal functioning problems related to treatment, and will also be affected by a complex interplay of aspects including emotional changes, body image, and partner interaction [7]. The majority of studies investigating radiation-induced vaginal morbidity are not recent and employed 2D planning/radiation methods. Compared to the modern treatment approaches such as IMRT and IG-IRT, at present, there has been an overall reduction in severe vaginal symptoms (for example, fistulas) when mild to moderate morbidity continues to be recorded in a significant proportion of patients [8].

Jensen et al. reported data collected between 1990 and 1993 on patients with cervix cancer who underwent EBRT plus IRT; considering the two-dimensional (2D) treatment planning used, approximately 85% of women had low or no sexual interest, 35% had moderate to severe lack of lubrication, 55% had mild to severe dyspareunia, and 30% were dissatisfied with their sex life. Reduced vaginal size was reported by 50% of patients and 45% were never, or only occasionally, capable of full intercourse. Despite sexual dysfunction and vaginal side effects, 63% of those sexually active before having cancer remained sexually active after treatment, albeit with a significantly reduced frequency [8].

Fibrosis and vaginal dryness are among the most important effects caused by a structural alteration of the tissues with a consequent increase in connective tissue, less elasticity, and a higher risk of developing vaginal fibrosis with the compromising of sexual health [9]. The study of sexual health remains a marginal topic and it is still considered a taboo subject in some cultures. Moreover, there is disagreement between patient and physician-reported symptom ratings, and combining the two assessments improves predicted accuracy [6,10].

The purpose of this work is to perform a comprehensive literature review evaluating the dysfunctions of the sexual sphere in patients with cervical cancer undergoing chemo-radiotherapy plus interventional radiotherapy.

## 2. Materials and Methods

### Search Strategy and Selection of Evidence

We performed a comprehensive literature search using Pub med, Scopus and Cochrane (up to December 2021) to find all full studies evaluating the dysfunctions of the sexual sphere in cervical cancer patients after chemo-radiation and interventional radiotherapy. ClinicalTrials.gov was searched for ongoing or recently completed trials, and PROSPERO was searched for ongoing or recently completed systematic reviews. In order to avoid missing relevant studies, we chose the following strategy burdened by high sensitivity and low specificity. An electronic search was supplemented by manually searching the references of included studies and review articles. The studies were identified using the following medical subject headings (MeSH) and keywords: “cervix cancer”, “surgery”, “external beam radiotherapy”, chemotherapy”, “brachytherapy”, “sexual dysfunction” and “toxicity”. The search strategy was: (“cervical neoplasm” [Mesh] OR “cervical neoplasm” [All fields]) AND (“brachytherapy” [Mesh] OR “brachytherapy” [All fields]), AND (“surgery” [Mesh] OR “surgery” [All fields]), AND (“chemotherapy” [Mesh] OR “chemotherapy” [All fields]), AND (“external beam radiotherapy” [Mesh] OR “external beam radiotherapy” [All fields]), AND “toxicity” [Mesh] OR “toxicity” [All fields]). The search was restricted to papers published in English. Only full-text clinical trials reporting on women with cervical cancer who had EBRT plus IRT and had sexual function measured using the female sexual functioning index were included (FSFI). The scale used to assess female sexual morbidity was the FSFI. It is a measure evaluating desire, arousal, orgasm, sexual pain and sexual satisfaction. It consists of 19 items scored according to a scale (0 or 1 to 5). Each subscale has a maximum score of six, resulting in a maximum overall score of 36 (indicative of better sexual function) and a score < 26 as cut off for diagnosis of sexual dysfunction. Conference papers, surveys, letters, editorials, book chapters, case reports, and reviews were excluded. A publication timeframe of 2000–2021 was considered. Studies were identified through a search process performed by three independent reviewers (FT, BF, and CC), and uncertainty regarding eligibility was resolved by a multidisciplinary committee (PC, CM, and VL). Eligible citations were retrieved for full-text review. An external expert committee defined the outcomes of benefit and harm (MF, GM). A multidisciplinary master board (CC, MAG, VV, LT, RI, RPDV, and AR) coordinated the project and performed the final independent check and the definitive approval of the review. The quality assessment showed high clinical and methodological heterogeneity and risks of bias in the included studies, making quantitative synthesis inappropriate. Therefore, meta-analysis outcomes were not reported.

## 3. Results

### 3.1. Studies Characteristics

The review was performed on 469 women included in six studies (five retrospective [11,12,13,14,15] and one prospective [6]) published from 2016 to 2021 (Figure 1). The median age was 46 (range 42–50 years). Data on interventional radiotherapy were available for two hundred seventy-nine (59.5%) patients. Three hundred fifteen patients received surgery (67%) [6,11,13,14,15]. Characteristics of the study are reported in Table 1.

### 3.2. Studies Description

The retrospective study of Bae et al. [11] evaluated 137 cervical cancer patients. Regarding treatment, 82 (62.0%) received surgery only, 22 (16.1%) received concurrent chemo-radiotherapy after surgery, 13 (9.5%) received concurrent chemotherapy and radiotherapy without surgery, 12 (8.8%) received radiotherapy after surgery, and 5 (3.6%) received chemotherapy after surgery. Interventional radiotherapy was performed on 40 patients. The participants experienced sexual dysfunction (4.83 ± 4.16). It was positively correlated with physical well-being, social well-being and functional well-being (*p* = 0.001). Patients who were 20–39 years old (*p* < 0.001), married (*p* = 0.031), had a high education (*p* = 0.022), worked (*p* < 0.001) and received CT treatment (*p* = 0.030) had considerably lower sexual activity [11].

The prospective study of Conway et al. [6] assessed patient-reported sexual adjustment, vaginal dosimetry and physician-reported vaginal toxicity in patients with cervical cancer treated with radio-chemotherapy (RCT) and MR-guided IRT. Patients completed the validated sexual adjustment questionnaire before IRT (baseline) and during follow-up. The diagnosis of cervical cancer and treatment negatively impacted sexual relationships in 61% and 39%, respectively. There were no significant changes in sexual adjustment over time (*p* = 0.599). There were no associations between sexual adjustment and the International Commission on Radiation Units and Measurements rectovaginal point dose or clinical vaginal involvement. Patients with higher International Federation of Gynaecologists and Obstetricians stages (≥IIB) had significantly worse sexual adjustment (*p* = 0.005). Desire significantly changed over time *(p* = 0.044) from the baseline pre-BT value with a temporary improvement from 3 to 6 months followed by a decrease at 9 months.

The retrospective study of Correia et al. [12] described the socio-demographic and clinical characteristics and those related to sexual life, and identified sexual dysfunction in women after cervical cancer treatment. Regarding treatment, 28 (60.8%) received concurrent chemo-radiotherapy, 9 (19.6%) received surgery and concurrent chemotherapy and 9 (19.6%) received surgery alone. Twenty-six patients received interventional radiotherapy. Out of a total of 46 women, 15 (32.61%) had sexual intercourse after treatment and 8 had an indication of sexual dysfunction (score 21.66; standard deviation = 7.06). The types of treatment (*p* = 0.03) and of radiotherapy (*p* = 0.01), in addition to the staging of the disease (*p* = 0.02), interfered with the sexual function. The most affected domains of the FSFI were lubrication (*p* = 0.03) and pain (*p* = 0.04).

Fakunle et al. [13] in a retrospective study reported the sexual function of 147 women treated for cervical cancer in the third, sixth and twelfth months after completing treatment. Eighty-nine patients received external beam radiotherapy and interventional radiotherapy, and fifty-eight patients received external beam radiotherapy, interventional radiotherapy and chemotherapy. The majority of the women (94.6%; n = 139) experienced sexual dysfunction, which persisted over time. The most affected domain was sexual arousal, whilst satisfaction was the least affected. Pain experienced during sexual activity persisted as time progressed. Age, educational level, the type of treatment received and having had sexual counseling before treatment commenced did not influence sexual function [13].

In the study of Frumovitz et al. [14], 114 patients (37 surgery, 37 radiotherapy and 40 controls) were included for analysis. When compared with surgery patients and controls using multivariate analysis, radiation patients had significantly poorer scores in sexual function. Univariate and multivariate analyses did not show significant differences between radical hysterectomy patients and controls on any of the outcome measures.

Finally, Zhou et al. [15] enrolled 140 patients with cervix cancer. A total of 57% (80/140) of the patients had undergone chemotherapy, and 108 of the 140 (77%) had radiotherapy. The prevalence of sexual dysfunction in the participants was 78%. Sexual function was affected by radiotherapy, age, type of surgery, sleep disorders and occupation.

Results are shown in Table 2.

### 3.3. Outcomes

Most studies showed that irradiated patients showed significantly worse sexual functioning. There were no significant changes in sexual adjustment over time [6,13]. The study by Fakunle et al. showed statistically significant differences between pain and time after completing treatment (*p* = 0.033) and satisfaction and the type of treatment received (*p* = 0.045) [13]. Zhou et al. showed that the main treatments responsible for sexual dysfunction were radiotherapy or radical hysterectomy [15]. Bae et al. reported worse pain and lubrication in patients who had undergone RT [11]; this study also showed that patients who were 20–39 years old (*p* < 0.001), married (*p* = 0.031), with high education (*p* = 0.022) and with work (*p* < 0.001) who underwent CT-based treatment (*p* = 0.030) had significantly worse sexual activity. Similar results were reported by Frumovitz et al. [14]. Indeed, in a multivariate analysis, irradiated patients showed significantly worse sexual functioning as measured by the FSFI. Conway et al. showed that patients with higher International Federation of Gynecologists and Obstetricians stages (>IIB) had significantly worse sexual adjustment (*p* = 0.005). However, the domain subset of desire significantly changed over time (*p* = 0.044) from the baseline pre-IRT value with a temporary improvement from 3 to 6 months, followed by a decrease at 9 months [6]. Finally, Correira et al. showed that surgery followed by radio-chemotherapy is associated with worse FSFI if compared to surgery alone [12].

### 3.4. Evidence to Decision Framework

In the cervix cancer setting, all therapeutic modalities available (surgery, external beam radiotherapy, interventional radiotherapy and chemotherapy) might interfere with sexual function. For the following reasons, the certainty of the evidence was deemed “very poor” for each outcome: indirectness for the population, including surgery, chemotherapy, and radiotherapy (EBRT, IRT), imprecision for sample size and, finally, possible selection bias due to a sub-group analysis.

## 4. Discussion

Sexual dysfunction in women after gynecological cancer treatment has not been very well analyzed in the literature. This current systematic review found that the sexuality sphere has only slightly been examined in patients with cervical cancer who are having surgery, chemo-radiation or interventional radiotherapy, despite the fact that sexual dysfunction and discomfort are a common treatment side effect [16,17,18,19,20,21,22]. Sexual dysfunction occurs more commonly than is reported in the literature [23], due to patients’ reluctance to address this problem, and lack of clinical time and economical resources to treat psychological, social and sexual aspects of patients’ illness experience [24]. Since studies have shown that vaginal toxicities can adversely affect the patient’s quality of life, physicians should discuss these issues with patients [25]. One might assume that vaginal changes would affect sexual function at least as much as the loss of a breast. The predominant interest in the breast is that breast cancer is more common than cancer of the female genital organs. Vaginal changes that may restrict coital pleasure include those that cause pain or bleeding during intercourse, those that interfere with the pleasure of feeling penile penetration and movement, and anatomical changes that interfere with achieving orgasm. There is no age above which sexual function was not important to women, and efforts to prevent vaginal changes, or to relieve them after therapy, should therefore be considered for women of all ages.

Despite the fact that chemo-radiation followed by IG-IRT plays an important role in the treatment of patients with cervical carcinoma, providing excellent local and pelvic control [3,4,5], it is linked to decreased sexual function, difficulty being sexually aroused, reaching orgasm, and obtaining sexual satisfaction when compared to women who have not received radiation treatment [12,14,15,26]. When comparing radiotherapy to surgery in early cervical cancer, a retrospective study by Cull et al. showed that it is correlated with more sexual issues than surgical treatment [27].

Morphological radiotherapy changes the lower vaginal epithelial volume and causes these sexual consequences. Patients who have undergone radiotherapy have a thinner epithelium, dense connective tissue, a low number of dermal papillae and a short distance from the basal layer to the epithelial surface with consequent atrophy [9]. The atrophic changes are considered to evolve over time where hypo-vascularization and hypoxia are predominant [28,29,30]. The dysfunctional elastin, fibrosis and low levels of circulating estrogen can explain the symptoms such as shortened, inelastic vagina and subsequent dyspareunia. These vaginal changes decrease lubrication and genital swelling during arousal and reductions in perceived vaginal length and elasticity during intercourse. The prevalence of dyspareunia is high among women with a history of cancer, and this disorder may be related to various changes in the vagina. Damage to peripheral nerves and small vessels as well as hormonal deficiencies and fibrosis may influence vaginal lubrication and genital swelling. Surgical removal of tissue, adhesions within the pelvis, and fibrosis may contribute to the perception of vaginal shortness and inelasticity. The long-term effects vary in individuals but are established after approximately two years when basal layer re-epithelialization and epithelial cell maturation are completed [29]. However, data on this topic are reported with an extremely variable incidence ranging from 1.2% to 88% [31]. Moreover, by destroying the oocytes and stopping estrogen synthesis, all the available treatments (radiotherapy, surgery and chemotherapy) cause low serum levels of estradiol, making the vagina less sensitive to systemic and topical estrogen [31]. Several studies have found that characteristics such as surgery, chemotherapy [20,32], age, marriage, high education, advanced stage, sleep difficulties and white-collar profession all predict increased sexual dysfunction [6,11,13,14,15].

Our systematic review showed contrasting results. Four out of six studies reported significantly worse sexual functioning in irradiated patients [12,13,14,15], while two showed no correlation between radiotherapy and sexual dysfunction [6,11]. We cannot disregard several factors that could explain the findings presented in the literature while studying them. Indeed, no details on radiotherapy treatment are reported in five studies [12,13,14,15]. Instead of more advanced radiotherapy procedures such as intensity modulated radiation therapy (IMRT) and magnetic resonance-based image-guided adaptive IRT (IGAIRT), patients were more likely treated with classic computed tomography-based 3-dimensional conformal irradiation. The EMBRACE study [32] found that severe vaginal morbidity was much lower following IGAIRT compared to data from studies employing previous techniques, and the majority of vaginal toxicity was mild or moderate (grade 1–2) [7]. Lindegaard et al., in 239 cervical cancer patients, showed an improvement in survival by approximately 15% and a reduction in moderate and severe morbidity by about 50% in women treated with MRI-based IGAIRT and IMRT chemo-radiation [33].

A review of the effects of surgery on sexual function found that there are different schools of thought regarding radical hysterectomy and its impact on sexual dysfunction. Some studies showed that sexual function declined initially, but they were not different from controls on long-term follow-up. Other studies concluded that patients who underwent radical hysterectomy had severe sexual dysfunction. Although scores improved over time, they never reached those of their healthy counterparts. Fertility-sparing/laparoscopic surgeries have outcomes comparable with open surgeries regarding sexual dysfunction [34,35,36,37]. Bergmark et al. published data on 256 Swedish cervical cancer survivors who had been treated with radical hysterectomy alone, radiation alone, and a combination of radiotherapy and surgery. They found no difference in lubrication, genital swelling/arousal, vaginal elasticity or length, or libido when comparing surgery patients with irradiated patients to patients who had received multiple-modality therapy. They did, however, find that the surgery-only patients reported statistically significant differences in vaginal lubrication, vaginal length, and vaginal elasticity when compared with controls [17]. In the present study, Zhou et al. showed that the main treatments responsible for sexual dysfunction were radiotherapy and radical hysterectomy [15]. Furthermore, the role of chemotherapy on vaginal toxicity could not be assessed [38,39]. Only the study of Bae et al. showed that patients who underwent CT-based treatment had significantly worse sexual activity.

Another thing worth mentioning is that sexual functions are linked to both physical and psychological sexual dysfunction. Despite the fact that vaginal difficulties in patients after irradiation are statistically reduced, several patients have poor sexual function despite normal vaginal function. After being diagnosed with cervical cancer, several patients lost their sexual desire [40,41]. Sexual activity loss or reduction occurs prior to therapy and is caused by vaginal pain and bleeding, vaginal discharge and emotional stress. The combination of psychological and physical symptoms of the condition leads to a loss of interest in sex and a reduction in sexual behaviors that lasts over time. The most difficult challenges in gynecological cancer patients are changes in women’s body image and sexual identity. Survivors experienced fear and anxiety about sexual activity [16,21,36,42], sexual competence [21] and future sexual encounters, as well as a lack of sexual activity [43]. Clinicians must identify patients who are most at risk for specific issues in order to give individualized therapies [41,42,43,44,45].

Some studies confirm that regular and long-term use of vaginal dilators/vibrators (or frequent penile–vaginal intercourse) will prevent or delay the development of vaginal adhesions and vaginal stenosis [46,47]. Many symptoms, including atrophy, dryness and pain, can be alleviated by restoring lubrication and an acid vaginal pH. Vaginal lubricating and moisturizing gels, hyaluronic acid, Vitamin A and E, and probiotics may reduce vaginal symptoms [17]. The application of local estrogen on vaginal tissues has been reported to be highly effective for the promotion of vaginal epithelial regeneration and consequently for reducing vaginal dryness [48]. Laser procedure has been reported as useful in small non-randomized series [49]. Websites were found to be useful and accessible as a first resource for information on sexual functioning after cancer [16]. Pelvic floor muscle strengthening and relaxation exercises are therefore recommended as well as psychosexual support to provide therapeutic rehabilitation and reassurance [50,51,52,53,54].

This research has a number of drawbacks. First, sample size imprecision, population indirectness, including surgery, chemotherapy and radiotherapy (external beam as well IRT), and the possible selection bias due to a sub-group analysis. Second, the final FSFI assessment and not the baseline was specified in four studies [11,12,14,15]. In the paper of Conway [6] and Fakunle [13], the FSFI assessment occurred before and at the end of the IRT treatment, respectively, not before EBRT, and this could have been impacted by the acute effects of EBRT.

Third, four out of six studies lack data on EBRT and IRT techniques and procedures, making it impossible to determine if the described dysfunctions might have been averted by using more contemporary treatments. Finally, just one old study (two-dimensional IRT) compared sexual function in patients receiving radiotherapy versus surgery showing that cervical cancer survivors who were treated with surgery alone can expect an overall quality of life and sexual function not unlike that of peers without a history of cancer.

Sexual function in patients treated for cancer should be considered from the beginning of treatment, and patients should be informed about possible treatment-related problems. Those patients who agreed to receive support should have available in the health care systems a team including their cancer therapists, psychologists, nurses and sexologists in order to increase their sexual function quality. Of course, every patient should be considered in a holistic concept adapted to their needs.

## 5. Conclusions

Based on these results, women who receive a diagnosis of cervix cancer should be informed and advised about sexual dysfunctions following treatment. Physical, physiological and social factors all contribute to the modification of the sexual sphere. Cervical cancer survivors who were irradiated have lower sexual and vaginal function than the normal population, but no consensus has been reached on the subject, mainly due to the inappropriate methodology of the studies. To address this issue, which is still overlooked today, randomized studies with a larger number of sexually active patients, a thorough assessment of sexual problems, and different methods of treatment are needed together with adequate patient information and decision participation.

## Figures and Tables

**Figure 1 medicina-58-01223-f001:**
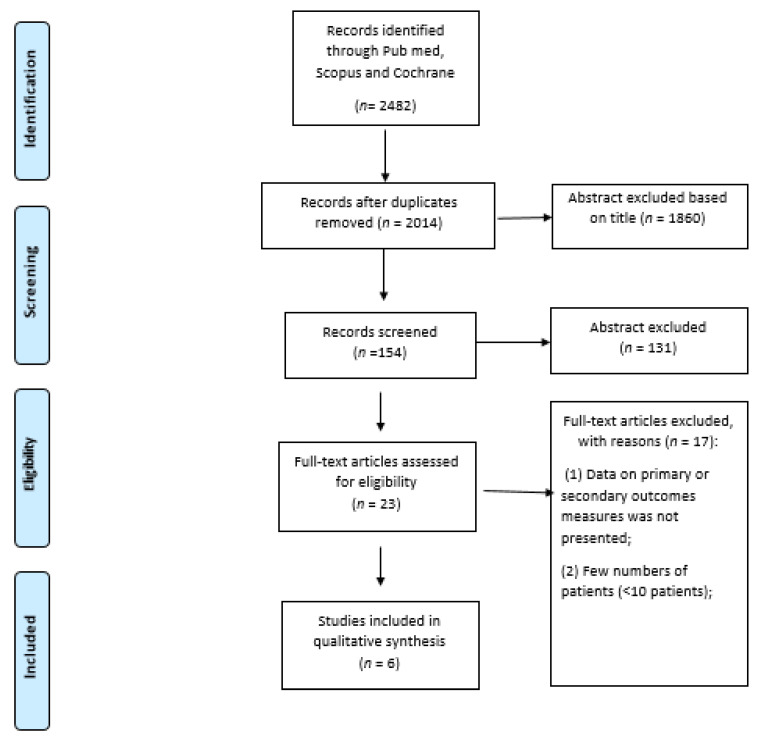
PRISMA Flowchart for outcomes and toxicity.

**Table 1 medicina-58-01223-t001:** Patient demographics, treatment characteristics, OS, LC, DFS, MSS and toxicity.

Author	Period	Study	Tumor	Sample Size, n	Median Age, Years	Stage		Type of Treatment	FSFI		QoL	Main Results
Bae [11]		Retrospective	Cervix	137	50 (28–59)	I: 114 II: 2 III: 1 IV: 2		Surgery: 85 Surgery + RT: 12 Surgery + RT-CT: 22 Surgery + CT: 5 RT-CT: 13	4.83 ± 4.16 Sexual desire: 1.53 ± 0.67 Satisfaction: 1.32 ± 0.64 Sexual arousal: 0.52 ± 0.88 Pain: 0.49 ± 0.86 Orgasm: 0.50 ± 0.82 Lubrication: 0.46 ± 0.75		FACT-G 57.33 ± 8.47 Physical well-being: 16.91 ± 3.21 Functional well-being: 4.47 ± 4.51 Emotional well-being: 14.00 ± 2.41 Social well-being: 11.95 ± 2.25	Cervical cancer patients with high sexual function tended to have low levels of depression and exhibited a higher quality of life
Frumovitz [14]	1991–98	Retrospective	Cervix	RT: 37 S: 37	RT: 46.9 S: 43.6	RT Ia1: 0 Ia2: 0 Ib1: 22 Ib2: 15	S Ia1: 1 Ia2:3 Ib1: 33 Ib2: 0	RT: 37 S: 37	RT: 17.1 Sexual desire: 2.9 Satisfaction: 3.2 Sexual arousal: 2.6 Pain: 2.7 Orgasm: 2.8 Lubrication: 2.9	S: 25.1 Sexual desire: 3.4 Satisfaction: 4.4 Sexual arousal: 4 Pain: 4.6 Orgasm: 4.2 Lubrication: 4.5	MCS RT: 47 S: 50.5 NS	Radiation patients had significantly poorer scores on health-related quality of life, psychosocial distress and sexual functioning. The disparity in sexual function remained significant in a multivariate analysis.
Conway [6]	2008–10	Retrospective	Cervix	27	50 (30–64)	IB: 15 IIA-IIB: 11 IIIA-B: 1		CRT + IRT: 27	Baseline: 16.4 1.5 months: 22.3 3 months: 23 6 months: 21.8 9 months: 23.9 12 months: 22.1			Patients with higher FIGO stages (≥IIB) had significantly worse sexual adjustment (*p* = 0.005)
Correira [12]	2015–16	Retrospective	Cervix	46	28 (30–49)			Surgery: 9 Surgery + RT: 9 RT-CT: 28 (IRT: 24)	Sexual desire: 3.20 Satisfaction: 4.23 Sexual arousal: 3.38 Pain: 4.10 Orgasm: 3.10 Lubrication: 3.65			The types of treatment (*p* = 0.03) and of RT (*p* = 0.01), and the stage (*p* = 0.02) interfered with the sexual function. The most affected domains of the FSFI were lubrication (*p* = 0.03) and pain (*p* = 0.04)
Fakunle [13]		Retrospective	Cervix	147	44 (30–73)			EBRT + IRT: 89 RT-CT + IRT: 58	Sexual desire: 2.5 Satisfaction: 3.1 Sexual arousal: 2.2 Pain: 2.4 Orgasm: 2.5 Lubrication: 2.4			Not find a statistically significant relationship between type of treatmentReceived, counseling, age and sexual function
Zhou [15]	2007–10	Retrospective	Cervix	140	45.6 (27–68)	I-III		Surgery: 120 CT: 80 RT: 108	Sexual desire: 2.58 Satisfaction: 2.80 Sexual arousal: 1.81 Pain: 2.47 Orgasm: 2.13 Lubrication: 2.09		FACT-G 124.45 Physical well-being: 22.2 ± 4.96 Functional well-being: 18.43 ± 5.65 Emotional well-being: 18.35 ± 4.75 Social well-being: 21.96 ± 4.60	Sexual function was affected by radiotherapy, age, type of surgery, sleep disorders, and occupation

Abbreviations: CT: chemotherapy; CRT: chemo-radiotherapy; FACT-G: Functional Assessment of Cancer Therapy–General Version; FIGO: International Federation of Gynecologists and Obstetricians; FSFI: Female Sexual Function Index; HADS: Hospital Anxiety and Depression Scale; ICRU: the International Commission on Radiation Units and Measurements; IRT: interventional radiotherapy; QoL: Quality of Life; RT: radiotherapy; S: surgery.

**Table 2 medicina-58-01223-t002:** Female sexual functioning index assessment.

Authors	Total FSFI	Desire	Arousal	Orgasm	Lubrification	Pain	Satisfaction	Main Results
Bae [11]	57.33 ± 8.47	1.53 ± 0.67	0.52 ± 0.88	0.50 ± 0.82	0.46 ± 0.75	0.49 ± 0.86	1.32 ± 1.64	Patients aged 20–39 years old (*p* < 0.001), married (*p* = 0.031), high education (*p* = 0.022), work (*p* > 0.001), spouse (*p* = 0.016) and CT treatment (*p* = 0.030) had significantly worse sexual activity
Conway [6]	Time 0: 48.5 12 m: 49.7	Time 0: 8.4 12 m: 8.9	Time 0: 2.8 12 m: 3.7	Time 0: 6.3 12 m: 6.4	Time 0: 7.8 12 m: 8.5	Time 0: 10 12 m: 7.3	Time 0: 6.4 12 m: 6.3	Patients with higher FIGO stages (>IIB) had significantly worse sexual adjustment (*p* = 0.005)
Correira [12]	21.66	3.20	3.38	3.10	3.65	4.10	4.23	The treatment interfered with sexual activity (*p* < 0.001). The type of treatment (*p* = 0.03), the type of RT (*p* = 0.01) and the stage of the disease (*p* = 0.02) were associated with whether or not the woman was sexually active after the treatment. Lubrication (*p* = 0.03) and pain (*p* = 0.04) were influenced by the type of treatment.
Fakunle [13]	At 3 m: 13.9 At 12 m: 15.7	At 3 m: 2.4 At 12 m: 2.5	At 3 m: 2.1 At 12 m: 2.2	At 3 m: 2.3 At 12 m: 2.5	At 3 m: 2.2 At 12 m: 2.4	At 3 m: 2.0 At 12 m: 2.8	At 3 m: 2.9 At 12 m: 3.2	In terms of the full FSFI score, only 5.4% (n = 8) of the women experienced good sexual function. Age, treatment received, and counseling did not impact sexual function
Frumovitz [14]	RT: 17.1 S + RT: 25.1	RT: 2.9 S + RT: 3.4	RT: 2.6 S + RT: 4.0	RT: 2.8 S + RT: 4.2	RT: 2.9 S + RT: 4.5	RT: 2.7 S + RT: 4.6	RT: 3.2 S + RT: 4.4	Irradiated patients showed significantly worse sexual functioning
Zhou [15]	13.9	2.58	1.81	2.13	2.09	2.47	2.80	Factors that were significantly associated with the total FSFI score were age, RT, sleep disorders, radical hysterectomy, and white-collar occupation (*p* < 0.05)

Abbreviation: CT: chemotherapy; FSFI: female sexual functioning index; FIGO: International Federation of Gynecologists and Obstetricians; m: months; RT: radiotherapy.

## Data Availability

Not applicable.

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
