# Peer review of "Assessment of Sexual Dysfunction in Cervical Cancer Patients after Different Treatment Modality: A Systematic Review"

_medicina, 2022, doi:10.3390/medicina58091223_

Round 1

Reviewer 1 Report

The topic of the work is very current due to the constantly growing number of cases of malignant neoplasms. Chemotherapy and radiotherapy are frequently used treatment methods  for women diagnosed with uterine cervix cancer. Side effects of these methods reduce the functioning of women in various spheres of life, including the sexual sphere, and thereby decline the overall quality of life. The presented study may be an inspiration to undertake further research on a similar subject, because it draws attention to a significant problem.

Author Response

Reviewer 1

Thank you very much for your appreciations

Reviewer 2 Report

1. The topic aims to review the sexual dysfunction in cervical cancer patients receiving chemo-radiation and brachytherapy. However, the patient populations in enrolled studies mainly underwent surgery. Among radiation therapy groups, the technique or treatment information of EBRT and/or brachytherapy was lacking. 

2. The Female Sexual Function Index adopt for evaluation of sexual dysfunction and assessment time were all different among enrolled studies. For the heterogeneous treatment factors including surgery, chemotherapy, EBRT, and brachytherapy and varied FSFI assessment in enrolled patients, it's hard to explore the correlation between sexual dysfunction and EBRT/brachytherapy via this review.

3. On page7, line195, the study might be Bae, et al(reference 11), not Correia, et al(reference 12)  to show age, married,... worse sexual activity.

4. On page8, line225-230, how to approve the correlation between thinner vaginal epithelium and sexual dysfunction? please give a more detailed description.

5. On page9, line269-270, "Vaginal lubricating and....probiotics" was an incomplete sentence.

Author Response

Reviewer 2

Moderate English changes required

As requested we revised the English

The topic aims to review the sexual dysfunction in cervical cancer patients receiving chemo-radiation and brachytherapy. However, the patient populations in enrolled studies mainly underwent surgery. Among radiation therapy groups, the technique or treatment information of EBRT and/or brachytherapy was lacking.

We agree with the reviewer. Unfortunately the analyzed works do not give information about EBRT and \or brachytherapy technique and treatment

The Female Sexual Function Index adopt for evaluation of sexual dysfunction and assessment time were all different among enrolled studies. For the heterogeneous treatment factors including surgery, chemotherapy, EBRT, and brachytherapy and varied FSFI assessment in enrolled patients, it's hard to explore the correlation between sexual dysfunction and EBRT/brachytherapy via this review

We completely agree with the reviewer. But despite these biases, the present review would like to provide an overview of toxicity after radio-chemotherapy and brachytherapy. Furthermore, as often underlined in the text, this work may be an inspiration to undertake further research on a similar subject, because it draws attention to a significant problem.

On page7, line195, the study might be Bae, et al(reference 11), not Correia, et al(reference 12)  to show age, married,... worse sexual activity

We changed the reference

On page 8, line225-230, how to approve the correlation between thinner vaginal epithelium and sexual dysfunction? please give a more detailed description.

As requested we give more detailed description

On page 9, line 269-270, "Vaginal lubricating and....probiotics" was an incomplete sentence.

We completed the sentence

Reviewer 3 Report

Comments to the Authors:

This manuscript evaluates the dysfunctions of the sexual sphere in patients with cervical cancer undergoing chemo-radiation and IRT. The results of this manuscript were mainly obtained by searching the relevant literatures on the Internet, and the literatures are complete. The analysis and studies of cervical cancer are very important and meaningful. This work is very novel and interesting. However, the following issues should be addressed .

1. The language should be improved. Some pictures are fuzzy, such as Figure 1, please provide a clearer one. Figure caption and figure need to be on one page. If the table is not on one page, please refer to the correct format.

2. Please elaborate the implications of evaluating the dysfunctions of the sexual sphere in patients with cervical cancer undergoing chemo-radiation and IRT. The article is like a data report based on references, without in-depth analysis and discussion, please supplement this part of the content

3. Articles cited are not of high quality. This may result in data and conclusions drawn from cited references having little reference value, which may lead to erroneous analysis in this paper. For example, reference 27 (IF = 1.85). It is recommended to refer to some high-level papers.

4. The authors stated that “Our systematic review showed contrasting results. Four out of 6 studies reported significantly worse sexual functioning in irradiated patients while two showed no correlation between radiotherapy and sexual dysfunction”. Why are there very different results, is the sample size imprecise or for other reasons? What analysis can be drawn from these different results?

5. The authors stated that “This research has a number of drawbacks”. Is there any way to overcome the drawbacks? Do these drawbacks affect the accuracy of the research?

6. The methods used in the article have some imperfections (mainly through online literature search). Some articles are aided by interviews and consultations of the clinical file. Please think about this question.

7. Introduction, the authors mentioned some effects of chemotherapy in cervical cancer patients. In order to support this statement, the following recently published important related papers should be cited: Chem. Soc. Rev. 2021, 50, 2839; Chem. Soc. Rev., 2017, 46, 7021.

Author Response

The language should be improved. Some pictures are fuzzy, such as Figure 1, please provide a clearer one. Figure caption and figure need to be on one page. If the table is not on one page, please refer to the correct format.

As requested we provaided English change. The figure and the tables are in one page

Please elaborate the implications of evaluating the dysfunctions of the sexual sphere in patients with cervical cancer undergoing chemo-radiation and IRT. The article is like a data report based on references, without in-depth analysis and discussion, please supplement this part of the content

Thank you for your suggestion. We supplemented this part of the content

Articles cited are not of high quality. This may result in data and conclusions drawn from cited references having little reference value, which may lead to erroneous analysis in this paper. For example, reference 27 (IF = 1.85). It is recommended to refer to some high-level papers.

We changed the reference with Support Care Cancer  IF 3.603

The authors stated that “Our systematic review showed contrasting results. Four out of 6 studies reported significantly worse sexual functioning in irradiated patients while two showed no correlation between radiotherapy and sexual dysfunction”. Why are there very different results, is the sample size imprecise or for other reasons? What analysis can be drawn from these different results?

We thanks the reviewer for this interesting question. We think that this different result may due by simple size, old radotherapy technique or lack on details on radiotherapy treatment used. Several studis showed as severe vaginal morbidity is much lower following intensity modulated radiotherapy and imagin guided interventional radiotherapy compared to data from studies employing older techniques. To overcome these problem, there should be more attention on the consequences that an important vaginal toxicity could bring, standardize radio-chemotherapy and interventional radiotherapy treatments and use advanced techniques. Naturally, randomized studies may be useful.

The authors stated that “This research has a number of drawbacks”. Is there any way to overcome the drawbacks? Do these drawbacks affect the accuracy of the research?

Very interesting question. The only way to overcome this drawbacks is draw up  randomized studies with a larger number of sexually active patients, a thorough assessment of sexual problems, and different methods of treatment. The retrospective studies have always bias that can affect the accuracy of the research

The methods used in the article have some imperfections (mainly through online literature search). Some articles are aided by interviews and consultations of the clinical file. Please think about this question.

We agree with the reviewer. In order to avoid the missing of relevant studies, we chose the following strategy burdened by high sensitivity and low specificity. 

Introduction, the authors mentioned some effects of chemotherapy in cervical cancer patients. In order to support this statement, the following recently published important related papers should be cited: Chem. Soc. Rev. 2021, 50, 2839; Chem. Soc. Rev., 2017, 46, 7021.

As requested we added the references suggested

Round 2

Reviewer 2 Report

For the discrepancy of enrolled patient populations and the topic of this review focus on patients receiving radiation therapy including brachytherapy, I suggest revising either the enrolled data or the topic.

Such as excluding reference 11(Bae H, et al), that surgery was the primary treatment among most patients, and only about 10% of patients received definitive radiation therapy if the author still wants to discuss the topic of radiation and correlated dysfunction.

Author Response

For the discrepancy of enrolled patient populations and the topic of this review focus on patients receiving radiation therapy including brachytherapy, I suggest revising either the enrolled data or the topic.Such as excluding reference 11( Bae H, et al), that surgery was the primary treatment among most patients, and only about 10% of patients received definitive radiation therapy if the author still wants to discuss the topic of radiation and correlated dysfunction.

we completely agree with the reviewer. As mentioned in the discussion, four out of six studies lack data on EBRT and IRT techniques and procedures, making it impossible to determine if the described dysfunctions might have been averted by using more contemporary treatments. Finally, just one old study (two-dimensional IRT) compared sexual function in patients receiving radiotherapy versus surgery showing that that cervical cancer survivors treated with surgery alone can expect overall quality of life and sexual function not unlike that of peers without a history of cancer. For this reason, we decided to change the paper focus as reported in the paper. Thank you very much for your suggestions

Reviewer 3 Report

  • The authors have fully addressed all the questions raised by the reviewer.

  •  

Author Response

The authors have fully addressed all the questions raised by the reviewer.

Thank you very much